# Variability in patient sociodemographics, clinical characteristics, and healthcare service utilization among 107,302 treatment seeking smokers in Ontario: A cross-sectional comparison

**Dolly Baliunas**[1,2]*, **Laurie Zawertailo**[1,3], **Sabrina Voci**[1], **Evgenia Gatov**[4], **Susan J. Bondy**[2], **Longdi Fu**[4], **Peter L. Selby**[1,2,5,6]

1 Addictions Program, Centre for Addiction and Mental Health, Toronto, Ontario, Canada, 2 Dalla Lana School of Public Health, University of Toronto, Toronto, Ontario, Canada, 3 Department of Pharmacology and Toxicology, Faculty of Medicine, University of Toronto, Toronto, Ontario, Canada, 4 ICES, Toronto, Ontario, Canada, 5 Department of Family and Community Medicine, University of Toronto, Toronto, Ontario, Canada, 6 Department of Psychiatry, University of Toronto, Toronto, Ontario, Canada

* dolly.baliunas@utoronto.ca

**Data Availability Statement:** The dataset from this study, composed of STOP program data and ICES datasets, is held securely in coded form at ICES.

## Abstract

### Background

Since 2005, the Smoking Treatment for Ontario Patients (STOP) program has provided smoking cessation treatment of varying form and intensity to smokers through 11 distinct treatment models, either in-person at partnering healthcare organizations or remotely via web or telephone. We aimed to characterize the patient populations reached by different treatment models.

### Methods

We linked self-report data to health administrative databases to describe sociodemographics, physical and mental health comorbidity, healthcare utilization and costs. Our sample consisted of 107,302 patients who enrolled between 18Oct2005 and 31Mar2016, across 11 models operational during different time periods.

### Results

Patient populations varied on sociodemographics, comorbidity burden, and healthcare usage. Enrollees in the Web-based model were youngest (median age: 39; IQR: 29–49), and enrollees in primary care-based Family Health Teams were oldest (median: 51; IQR: 40–60). Chronic Obstructive Pulmonary Disease and hypertension were the most common physical health comorbidities, twice as prevalent in Family Health Teams (32.3% and 30.8%) than in the direct-to-smoker (Web and Telephone) and Pharmacy models (13.5%-16.7% and 14.7%-17.7%). Depression, the most prevalent mental health diagnosis, was twice as prevalent in the Addiction Agency (52.1%) versus the Telephone model (25.3%).

While data sharing agreements and privacy legislation for the province of Ontario prohibit ICES from making the dataset publicly available, access may be granted to those who meet pre-specified criteria for confidential access, available at https://www.ices.on.ca/DAS. Requests to access ICES data may be submitted to ICES Data & Analytic Services at das@ices.on.ca, with information available at https://www.ices.on.ca/DAS/Submitting-your-request.

**Funding:** Funding for this work was provided by a grant from the Ontario Ministry of Health and Long Term Care (http://www.health.gov.on.ca/en/) Health Service Research Fund (#430) to DB and PLS (Co-Is). Funding for STOP was provided by the Ontario Ministry of Health and Long Term Care (#HLTC5047FL) as part of the Smoke Free Ontario Strategy to PLS and LZ (Co-PIs). The funders had no role in study design, data collection and analysis, decision to publish, or preparation of the manuscript. PLS would like to acknowledge salary support for his clinician-scientist position from the Centre for Addiction and Mental Health (www.camh.ca) and the Department of Family and Community Medicine at the University of Toronto (https://www.dfcm.utoronto.ca/).

**Competing interests:** I have read the journal's policy and the authors of this manuscript have the following competing interests: DB reports grants from the Canadian Cancer Society Research Institute during the conduct of the study; she also reports grants from a Pfizer GRAND award, outside the submitted work. LZ reports grants from the Canadian Cancer Society Research Institute during the conduct of the study; she also reports grants from Pfizer GRAND Awards, outside the submitted work. PLS reports receiving grants from the Centre for Addiction and Mental Health, Ontario Ministry of Health and Long-Term Care, Health Canada, Canadian Institutes of Health Research, Canadian Cancer Society Research Institute, Canadian Partnership Against Cancer, Medical Psychiatry Alliance, Ontario Neurotrauma Foundation, the Patient-Centered Outcomes Research Institute, Bhasin Consulting Fund Inc and Pfizer Inc./Canada. PLS also reports serving on advisory boards and receiving consulting fees and honoraria from Pfizer Canada Inc., Johnson & Johnson Group of Companies, Evidera Inc., Miller Medical Communications, NVision Insight Group, Myelin & Associates, University of Ottawa Heart Institute, Royal College of Physicians and Surgeons of Canada, Royal Victoria Regional Health Centre, the University of Toronto Department of Family and Community Medicine, Northern Ontario School of Medicine, Canadian Partnership Against Cancer,

Median healthcare costs in the two years up to enrollment ranged from $1,787 in the Telephone model to $9,393 in the Addiction Agency model.

## Discussion

While practitioner-mediated models in specialized and primary care settings reached smokers with more complex healthcare needs, alternative settings appear better suited to reach younger smokers before such comorbidities develop. Although Web and Telephone models were expected to have fewer barriers to access, they reached a lower proportion of patients in rural areas and of lower socioeconomic status. Findings suggest that in addition to population-based strategies, embedding smoking cessation treatment into existing healthcare settings that reach patient populations with varying disparities may enhance equitable access to treatment.

## Introduction

Globally, more than 1 billion people smoke tobacco [1]. The World Health Organization Framework Convention on Tobacco Control (WHO FCTC) has endorsed six tobacco control policies to address the global tobacco epidemic: monitor tobacco use and prevention policies; protect people from tobacco use; warn about the dangers of tobacco; enforce bans on tobacco advertising, promotion and sponsorship; raise taxes on tobacco; and offer help to quit tobacco use [1]. Canada is a party to the WHO FCTC, and the province of Ontario has implemented the six endorsed policies since 2005.

Approximately 2 million people in Ontario currently smoke cigarettes [2], and the province spends more than $2 billion dollars annually in direct healthcare costs to treat smoking-related disease [3]. Each year, almost half of Ontarians who smoke try to quit [4], but without help few quit attempts lead to long-term abstinence [5]. Evidence-based treatments can improve quit rates. For example, there is extensive and high-quality evidence that nicotine replacement therapy (NRT) increases the likelihood of quitting smoking by 50% to 60%, irrespective of treatment setting [6]. Further, there is evidence that combining counselling with NRT is more effective than providing either alone [6, 7]. Offering NRT for free or at a reduced cost has been shown to increase the proportion of smokers who make a quit attempt, use smoking cessation treatment, and successfully quit smoking [8–12]. Based on this evidence, the Smoking Treatment for Ontario Patients (STOP) program was established in 2005, with funding from the government of Ontario, to provide NRT and behavioural support at no cost to Ontarians wanting to quit smoking in a variety of treatment settings. The STOP program has delivered smoking cessation treatment in 11 distinct models, which intentionally vary in recruitment methods, point of contact, mode of delivery, personnel of delivery, intensity of contact and combinations of behavioural support and pharmacotherapies (see Table 1). Models include those engaging people who smoke both directly and through partnering healthcare organizations [13–17], via either self-referral or referral from a healthcare practitioner. Since its inception, STOP has utilized a combination of both clinic-based and population-based treatment models. Population-based strategies have been used to distribute a standard supply of NRT with brief counselling, which has included: mail-out of NRT to individuals who enrolled via telephone or website; providing eligible smokers with vouchers for NRT to be redeemed at a local partnering pharmacy; and distributing NRT kits at smoking cessation workshops held at

Battle River Treaty 6 Healthcare, Lung Association of Nova Scotia, Exchange Summit, Toronto Public Health, Ontario Association of Public Health Dentistry and ECHO. PLS also reports that MedPlan Communications assisted in organizing Pfizer Canada Inc. Advisory Board events for which he was a consultant and that Pfizer Inc., Novartis, and Johnson & Johnson are vendors of record for providing free/discounted smoking cessation pharmacotherapy for research studies on which he is principal or co-investigator. No other author has competing interests to declare. This does not alter our adherence to PLOS ONE policies on sharing data and materials.

local Public Health Units. Clinic-based models have operated in various primary and specialized care settings across the province, including addictions treatment agencies; treatment in these settings evolved over time from standardized to more personalized in order to provide patients with more intensive pharmacological and behavioural support, as needed.

In addition to increasing the number of quits and quit attempts, the overall goal of the STOP program was also to provide more equitable access to smoking cessation services by reducing geographic and financial barriers to treatment. Smoking is unevenly distributed in the population, with higher prevalence among disadvantaged groups [18–20]. Given evidence that smoking cessation programs may widen inequalities in smoking [21] by reaching easier to access and more advantaged populations [22], our objective was to identify and characterize any variation in the population of treatment seeking smokers reached by the different treatment models offered by STOP. Although patients who access the STOP program complete a self-reported baseline survey, data linkage across health administrative datasets provides a unique opportunity to understand who patients are in terms of medical and mental health comorbidity and how they interact with the healthcare system. Combining these data with self-reported measures (e.g., employment status, smoking behaviour), not typically available in healthcare administrative datasets, allows for a picture of the patient population of unprecedented depth and breadth and can influence health policy decisions in Ontario and beyond.

## Methods

### Study design and population

We conducted a descriptive cross-sectional study of all individuals who sought smoking cessation treatment via the STOP program in Ontario, Canada using self-reported baseline assessment data linked to health administrative data held at ICES (https://www.ices.on.ca/). ICES is an independent, non-profit research institute whose legal status under Ontario's health information privacy law allows it to hold and use administrative, population health, clinical and other data files for the purposes of analysis, evaluation, and decision support.

We included all individuals aged 18 to 105 years, who enrolled in the STOP program between 18 October 2005 and 31 March 2016. We excluded individuals whose records could not be linked across datasets due to express denial of consent to linkage, missing or invalid linkage information, non-Ontario residence, and records with data inconsistencies (e.g., death date preceding STOP enrollment). The index date was defined as date of enrollment in the STOP program. For individuals who enrolled more than once within the study timeframe, only data related to their first enrollment was used. The 11 treatment delivery models developed and implemented in STOP are described briefly in Table 1; models varied on treatment offered, setting, type and degree of patient contact, patient eligibility criteria, and years of operation. Individuals who enrolled in these models form the comparison groups for this study.

### Data sources

The STOP program patient-level enrollment and follow-up survey data were linked to ICES data holdings using a combination of probabilistic and deterministic linkage with a 96% linkage rate. Datasets were linked by means of unique encoded identifiers and analyzed at ICES. To capture sociodemographic information (including age, sex and postal code) for all Ontarians eligible for health coverage, we used the Registered Persons Database [23], a central population registry file which enables linkage across population-based health administrative datasets. We used the 2006 Statistics Canada census [24] to link residential postal code to neighbourhood income quintile; the Immigration, Refugee and Citizenship Canada Permanent Resident database [25] to identify immigration category; and the Rurality Index of

**Table 1. Smoking cessation treatment model characteristics.**

| Treatment model | Timeframe | No. (%) | No. sites | Patient contact | Treatment offered | | Patient eligibility |
|---|---|---|---|---|---|---|---|
| | | | | | Behavioural support | NRT | |
| **Practitioner-mediated** | | | | | | | |
| Family Health Team (FHT) | 2011–2016 | 32,618 (30.4) | 209 | ▪ Visits with a healthcare provider[a] | ▪ Individual and/or group | ▪ Individualized | ▪ Rostered patients |
| | | | | ▪ Frequency of visits not prescribed by STOP but recommended every 2–4 weeks | ▪ Intensity (number of sessions and minutes per session) not prescribed by STOP | ▪ Up to 26 weeks of NRT within 1 year (max of 4 weeks at one visit) | ▪ Cigarette smokers wanting to quit, reduce or maintain recent abstinence from smoking |
| | | | | | | ▪ Patches and/or short-acting | |
| Nurse Practitioner-Led Clinic (NPLC) | 2014–2016 | 566 (0.5) | 20 | | | ▪ Dosing as per clinical discretion (with medical directive) | |
| Community Health Centre-B (CHC-B) | 2012–2016 | 5,862 (5.5) | 59 | | | ▪ May provide standard kit | |
| Addiction Agency (AA) | 2012–2016 | 3,408 (3.2) | 49 | | | ▪ Some Addiction Agency sites arrange mail out of 10-week NRT kit following in-person assessment | |
| Tertiary care (hospital-based) | 2005–2009 | 1,672 (1.6) | 3 | ▪ Visits with a healthcare provider[a] or a smoking cessation specialist | ▪ Intensity and frequency of counselling varied, and some patients did not receive counselling | ▪ Standard kits | ▪ Adults (18+ years) |
| | | | | | | ▪ Up to 10 weeks | ▪ Daily smoking 10 + cigarettes |
| | | | | | | ▪ Patch or short-acting | ▪ Plan quit attempt within 30 days |
| | | | | | | ▪ On-label use | ▪ No contraindications to NRT[b] |
| Community Health Centre-A (CHC-A) | 2007–2009 | 401 (0.4) | 14 | ▪ Visits with a healthcare provider[a] | ▪ 3 sessions of brief individual cessation counselling | ▪ Standard kits | |
| | | | | | | ▪ Up to 10 weeks | |
| | | | | | | ▪ Patch or short-acting | |
| | | | | | | ▪ On-label use | |
| Workshop | 2007–2016 | 18,539 (17.3) | 58 | ▪ Group workshop | ▪ 1-hour standard psychoeducation presentation | ▪ Standard kit | |
| | | | | ▪ Brief one-on-one consultation with Public Health Unit or STOP staff | | ▪ 5 or 10 weeks | |
| | | | | | | ▪ Patch and/or short-acting | |
| | | | | | | ▪ On-label use | |
| Public Health Unit (PHU) | 2006–2008 | 1,509 (1.4) | 12 | ▪ Visits with Public Health Unit staff | ▪ Patient may be referred to optional individual or group counselling if available | ▪ Standard kit | |
| | | | | | ▪ Type of counselling varied and was determined by site | ▪ Up to 10 weeks | |
| | | | | | | ▪ Patch or short-acting | |
| | | | | | | ▪ On-label use | |
| Pharmacy (community) | 2007–2008 | 6,412 (6.0) | 98 | ▪ Web-based enrollment | ▪ Brief (5–10 mins) individual counselling | ▪ Standard kits | |
| | | | | ▪ 1–3 visits with pharmacist | ▪ Randomized to 1 or 3 sessions | ▪ Up to 5 weeks | |
| | | | | | | ▪ Patch or short-acting | |
| | | | | | | ▪ On-label use | |

(*Continued*)

**Table 1.** (Continued)

| Treatment model | Timeframe | No. (%) | No. sites | Patient contact | Treatment offered | | Patient eligibility |
|---|---|---|---|---|---|---|---|
| | | | | | **Behavioural support** | **NRT** | |
| **Direct-to-smoker** | | | | | | | |
| Web | 2008–2009; 2011 | 6,748 (6.3) | NA | ▪ Web-based enrollment | ▪ Self-help materials | ▪ Standard kit sent by mail | ▪ Adults (18+ years) |
| | | | | | | ▪ 5 weeks | ▪ Daily smoking 10 + cigarettes |
| | | | | | | ▪ Patch or short-acting | ▪ Plan quit attempt within 30 days |
| | | | | | | ▪ On-label use | ▪ No contraindications to NRT[b] |
| Telephone | 2006–2010 | 29,567 (27.6) | NA | ▪ Telephone call with call centre agent or STOP research staff | ▪ Self-help materials | ▪ Standard kit sent by mail | |
| | | | | | ▪ 2009: 5 motivational phone messages | ▪ 5 to 10 weeks | |
| | | | | | | ▪ Patch or short-acting | |
| | | | | | | ▪ On-label use | |

NA = not applicable; NRT = nicotine replacement therapy.

[a] Healthcare providers include physician, nurse practitioner, registered nurse, registered practical nurse, pharmacist, respiratory educator or therapist, mental health counsellor, addiction counsellor, social worker, health promoter.

[b] Contraindications to NRT as listed on product monograph at the time of treatment

Ontario [26] to link postal code to rurality of the community. The following validated prevalent morbidity cohorts were used to determine medical comorbidity: asthma, coronary heart failure, chronic obstructive pulmonary disease (COPD), diabetes mellitus, myocardial infarction (MI), and hypertension (S1 Appendix). We further ascertained presence of a prior cancer diagnosis using the Ontario Cancer Registry [27]. To examine healthcare service utilization, medical hospitalizations were identified through the Canadian Institute for Health Information Discharge Abstract Database [28]. Psychiatric hospitalizations were ascertained using the Ontario Mental Health Reporting System [29], which contains information on all designated adult psychiatric beds in Ontario. Outpatient physician visits were captured through the Ontario Health Insurance Plan database [30], which contains claims for services billed by Ontario physicians. Emergency department (ED) visits were captured through the National Ambulatory Care Reporting System [31]. Lastly, to ascertain health system costs, several additional datasets were used, as described elsewhere (S2 Appendix) [32, 33].

## Outcomes

For each individual, descriptive characteristics were measured at index. The following characteristics were derived by self-report from the STOP baseline assessment: employment status, educational attainment, smoking behaviours (number of cigarettes per day, time to first cigarette), and prevalent mental health comorbidities (anxiety, bipolar disorder, depression, schizophrenia). Additional characteristics were derived via linkage to ICES datasets: age, sex, rurality, neighbourhood income quintile, immigrant category, and prevalent medical comorbidities.

Healthcare service utilization and cost outcomes were measured for the period of 2 years up to and including the index date. Healthcare service utilization measures of interest were any visit and median number of visits of the following type: primary care outpatient visits;

specialist outpatient visits (overall and mental health and addictions related); ED visits and hospitalizations (overall and mental health and addictions related). For additional details regarding mental health and addictions related outcome definitions, see S3 Appendix. Total healthcare costs were derived by applying a person-centred costing approach using a well-established methodology [32] that uses comprehensive healthcare costs from administrative data for all major sectors of healthcare spending: inpatient hospitalizations, physician visits, complex continuing care, long-term care, home services, assistive devices and pharmaceuticals (for certain individuals, e.g. seniors).

## Analysis

For each STOP model, we described the number and proportion of individuals with regard to each of the sociodemographic, clinical, and healthcare service use characteristics at the time of index enrollment. The proportion of missing data on individual characteristics was calculated for each model separately; less than 5% missing data on a characteristic was ignored; 5–10% missing data on a characteristic was noted with a symbol in the relevant table; and where greater than 10% of data was missing, the characteristic was not reported for that model. To examine healthcare service utilization, we ascertained the number and proportion of individuals who had each type of service encounter in the 2 years prior to index, and generated the median number of visits per person. For healthcare costs 2 years prior and including the index, we report median overall cost per person. Statistical analyses were conducted using SAS version 9.4.

## Ethics

Each model of the STOP program, as well as this analysis and data linkage, was approved by the Research Ethics Board of the Centre for Addiction and Mental Health (#027/2016, #081/2005, #015/2006, #064/2006, #223/2006, #281/2006, #128/2007, #264/2008, #058/2011, #154/2012). Patients in each model provided written informed consent, with the exception of the Telephone model, where consent was provided over telephone either verbally to a live agent or via an interactive voice response system and recorded in a database.

## Results

### Study population

Of 132,509 STOP enrollment records between 18 October 2005 and 31 March 2016, we excluded 5,863 records that could not be linked across datasets or had data inconsistencies, and 679 records of non-Ontario residents. We then selected the first enrollment per person, resulting in a cohort of 107,828 unique individuals. After further excluding 526 persons not eligible for provincial health coverage, the final cohort was N = 107,302. Approximately one-third (30.4%) of patients were enrolled in the Family Health Team model, another quarter (27.6%) were enrolled in the Telephone model, 17.3% were enrolled in Workshops, and the rest were distributed across other models.

### Sociodemographic and smoking-related characteristics

Patients were not a homogeneous group; there was variation across STOP models with regard to individual characteristics. Enrollees in the Web model had the youngest (39 IQR 29–49) and Family Health Teams the oldest (51 IQR 40–60) median age (Fig 1). Nine of eleven models had a little more than 50% female patients; Addiction Agencies had the lowest proportion of female patients (39.0%). The Family Health Team, Community Health Centre-A, and Public

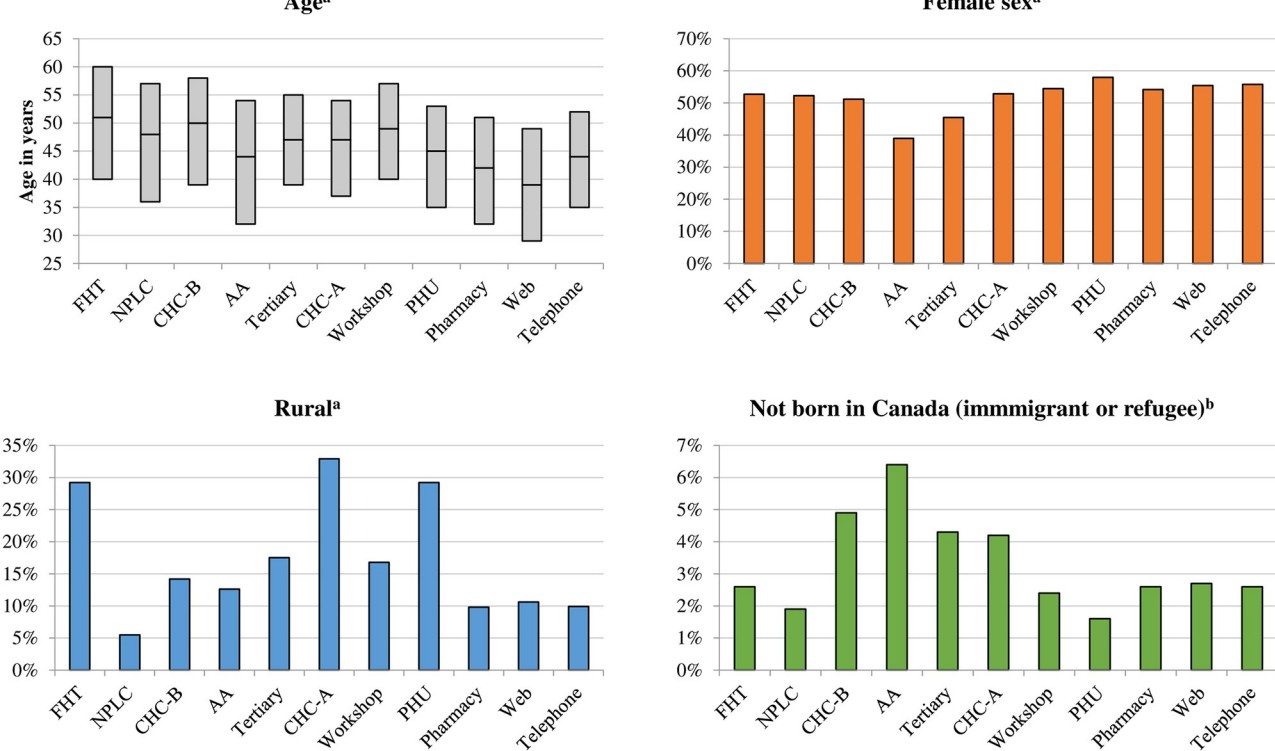

**Fig 1. Select demographic characteristics, by STOP model.** Model (years of operation): FHT = Family Health Team (2011–2016); NPLC = Nurse Practitioner-Led Clinic (2014–2016); CHC = Community Health Centre (A, 2007–2009; B, 2012–2016); AA = Addiction Agency (2012–2016); Tertiary care (2005–2009); Workshop (2007–2016); PHU = Public Health Unit (2006–2008); Pharmacy (2007–2008); Web (2008–2009; 2011); Telephone (2006–2010). [a]Ascertained from the Registered Persons Database. [b]Ascertained using the Immigration, Refugee and Citizenship Canada Permanent Resident database (immigrant and refugee categories were combined to minimize risk of re-identification due to small cell sizes).

Health Unit models had the highest concentrations of rural enrollees (29.2% to 32.9%) and the Nurse Practitioner-Led Clinics had the lowest (5.5%), followed by the Pharmacy, Web and Telephone models (9.8% to 10.6%). Addiction Agencies had the highest concentration of immigrant and refugee patients (6.4%). The highest proportion of unemployed enrollees was in Addiction Agencies (64.1%) and the lowest in the Pharmacy, Web and Telephone models (<40%, Fig 2). The highest proportion of individuals living in the lowest-income neighbourhood areas were in the Nurse Practitioner-Led Clinic (44.2%) and Community Health Centre-B models (43.0%); the lowest proportion was in Family Health Teams (27.8%). The highest proportion of patients who did not complete high school were in Community Health Centres (≥33%). Lastly, the proportion of patients who self-reported smoking 20+ cigarettes per day and smoking their first cigarette within 5 minutes of waking was highest in the Tertiary Care model (Fig 3).

## Comorbid conditions, healthcare service utilization and costs

COPD and hypertension were the most common medical comorbidities across STOP models and most prevalent (about one-third of patients) in Family Health Teams (Table 2). Self-reported mental health comorbidities were prevalent across all models, but most highly prevalent in the Nurse Practitioner-Led Clinic and Addiction Agency models with over 50% of patients reporting lifetime diagnoses by a healthcare provider of anxiety or depression. Direct-to-smoker (Web and Telephone) and Pharmacy models had the lowest comorbidity burden.

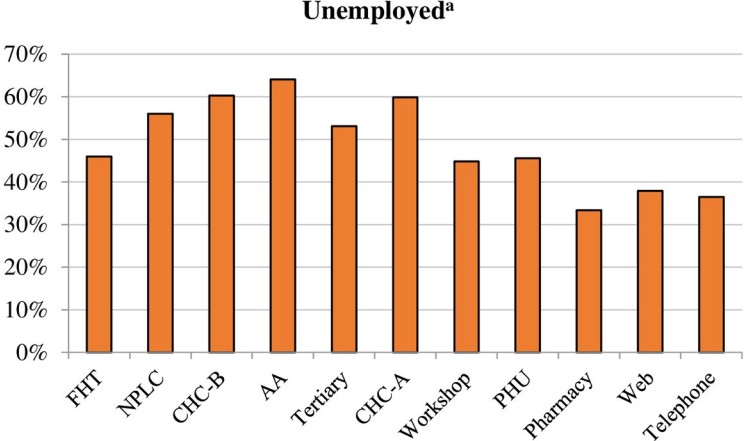

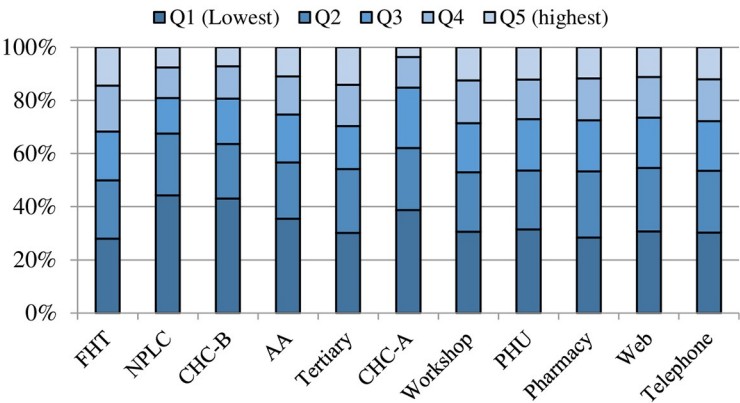

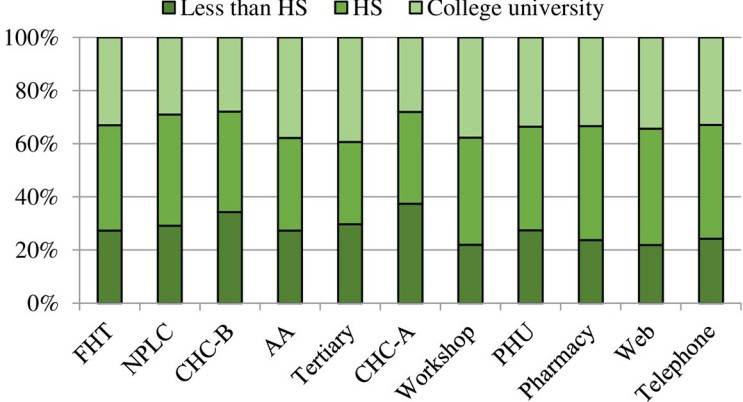

**Fig 2. Select socioeconomic characteristics, by STOP model.** Model (years of operation): FHT = Family Health Team (2011–2016); NPLC = Nurse Practitioner-Led Clinic (2014–2016); CHC = Community Health Centre (A, 2007–2009; B, 2012–2016); AA = Addiction Agency (2012–2016); Tertiary care (2005–2009); Workshop (2007–2016); PHU = Public Health Unit (2006–2008); Pharmacy (2007–2008); Web (2008–2009; 2011); Telephone (2006–2010). [a]Ascertained using patient self-report in STOP baseline survey. [b]Ascertained using postal code information linked to 2006 census data.

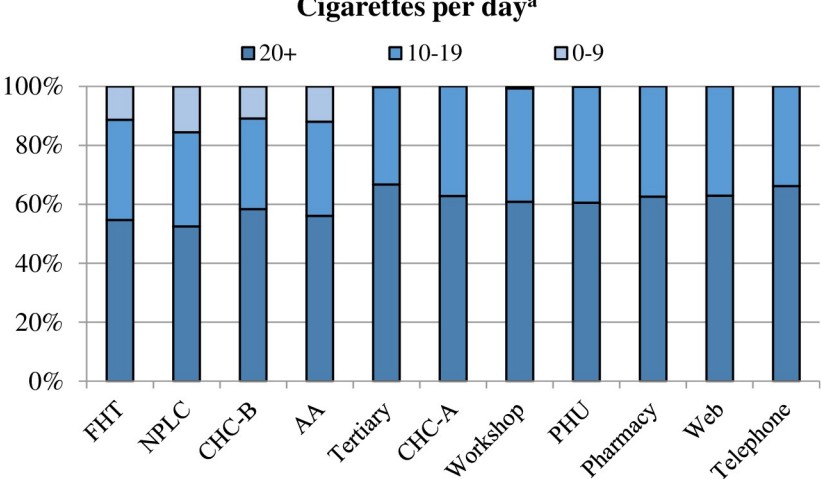

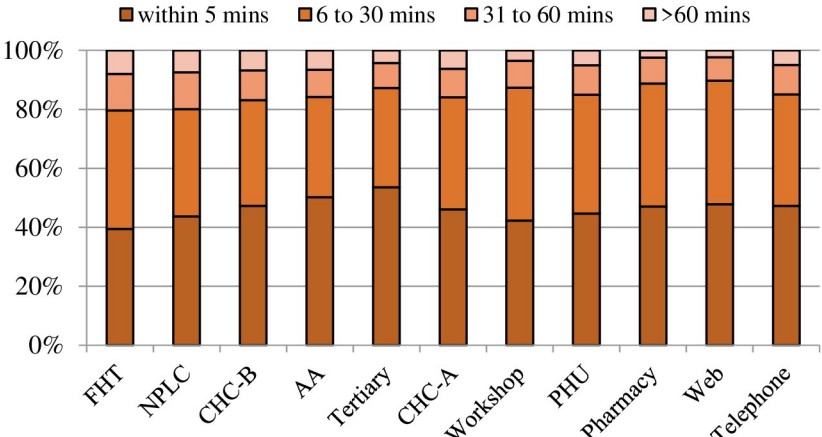

**Fig 3. Select smoking-related characteristics, by STOP model.** Model (years of operation): FHT = Family Health Team (2011–2016); NPLC = Nurse Practitioner-Led Clinic (2014–2016); CHC = Community Health Centre (A, 2007–2009; B, 2012–2016); AA = Addiction Agency (2012–2016); Tertiary care (2005–2009); Workshop (2007–2016); PHU = Public Health Unit (2006–2008); Pharmacy (2007–2008); Web (2008–2009; 2011); Telephone (2006–2010). [a]Ascertained using patient self-report in STOP baseline survey.

The proportion of STOP patients who saw a family physician or general practitioner for primary care in the 2 years prior to enrollment was high across all but the Community Health Centre and Nurse Practitioner-Led Clinic models, where such care is more often provided by other practitioner types (Table 3). Addiction Agency and Tertiary Care patients visited primary care providers the most, with a median of 11 visits during this period. Medical hospitalizations were most common among Tertiary Care patients. Addiction Agency patients had the greatest prevalence of outpatient physician visits to a psychiatrist or a primary care provider for mental health and addictions related reasons. However, patients treated for smoking cessation at an Addiction Agency also had the greatest prevalence of specialist physician visits and ED use, and the second greatest prevalence of hospitalizations; with only one third of ED visits and one quarter of hospitalizations being due to mental health and addictions concerns, the vast majority were for physical comorbidities. Lastly, Addiction Agency patients incurred the

**Table 2. Patient comorbidities at time of enrollment, by STOP model.**

| Model | FHT | NPLC | CHC-B | AA | Tertiary | CHC-A | Workshop | PHU | Pharmacy | Web | Telephone |
|---|---|---|---|---|---|---|---|---|---|---|---|
| | N = 32,618 | N = 566 | N = 5,862 | N = 3,408 | N = 1,672 | N = 401 | N = 18,539 | N = 1,509 | N = 6,412 | N = 6,748 | N = 29,567 |
| Medical comorbidities derived from health administrative data linkage. No. (%) | | | | | | | | | | | |
| Asthma | 3,059 (9.4) | 57 (10.1) | 655 (11.2) | 433 (12.7) | 155 (9.3) | 32 (8.0) | 1,686 (9.1) | 129 (8.5) | 428 (6.7) | 616 (9.1) | 2,283 (7.7) |
| Cancer | 1,537 (4.7) | 17 (3.0) | 222 (3.8) | 85 (2.5) | 82 (4.9) | 17 (4.2) | 724 (3.9) | 49 (3.2) | 182 (2.8) | 155 (2.3) | 927 (3.1) |
| CHF | 966 (3.0) | 14 (2.5) | 163 (2.8) | 57 (1.7) | 64 (3.8) | † | 350 (1.9) | 22 (1.5) | 59 (0.9) | 81 (1.2) | 281 (1.0) |
| COPD | 10,544 (32.3) | 142 (25.1) | 1,616 (27.6) | 813 (23.9) | 419 (25.1) | 62 (15.5) | 4,787 (25.8) | 310 (20.5) | 877 (13.7) | 914 (13.5) | 4,938 (16.7) |
| Diabetes | 5,592 (17.1) | 80 (14.1) | 933 (15.9) | 443 (13.0) | 246 (14.7) | 42 (10.5) | 2,412 (13.0) | 171 (11.3) | 514 (8.0) | 547 (8.1) | 2,294 (7.8) |
| Hypertension | 10,038 (30.8) | 133 (23.5) | 1,386 (23.6) | 701 (20.6) | 440 (26.3) | 58 (14.5) | 4,699 (25.3) | 277 (18.4) | 1,083 (16.9) | 992 (14.7) | 5,245 (17.7) |
| MI | 1,280 (3.9) | 17 (3.0) | 201 (3.4) | 60 (1.8) | 129 (7.7) | † | 503 (2.7) | 35 (2.3) | 78 (1.2) | 93 (1.4) | 339 (1.1) |
| Mental health comorbidities derived from STOP survey self-report. No. (%) | | | | | | | | | | | |
| Anxiety | 10,110 (31.0) | 314 (55.5) | 2,516 (42.9) | 1,682 (49.4)* | 507 (30.3) | 141 (35.2) | 5,420 (29.2) | 440 (29.2) | ** | ** | 5,806 (19.6) |
| Bipolar | 1,406 (4.3) | 58 (10.2) | 557 (9.5) | 453 (13.3)* | 112 (6.7) | 29 (7.2) | 996 (5.4) | 93 (6.2) | ** | ** | 1,050 (3.6) |
| Depression | 11,676 (35.8) | 298 (52.7) | 2,827 (48.2) | 1,775 (52.1) | 616 (36.8) | 161 (40.1) | 6,303 (34.0) | 567 (37.6) | ** | ** | 7,481 (25.3) |
| Schizophrenia | 608 (1.9) | 48 (8.5) | 337 (5.7) | 297 (8.7)* | 66 (3.9) | 18 (4.5) | 513 (2.8) | 51 (3.4) | ** | ** | 459 (1.6) |

Percent (%) observed are within valid data (missing data ignored) unless otherwise indicated; AA = Addiction Agency, CHC = Community Health Centre, FHT = Family Health Team, NPLC = Nurse Practitioner-Led Clinic, PHU = Public Health Unit, CHF = coronary heart failure, COPD = chronic obstructive pulmonary disease, MI = myocardial infarction;

* proportion of missing 5–10%;

** proportion of missing >10%;

† suppressed due to small cell counts (<6).

most health system costs (median $9,393) during the 2 years prior to STOP enrollment, with a median cost five times higher than Telephone patients who incurred the least cost (median $1,787).

## Discussion

We compared the characteristics of patients who sought smoking cessation treatment via 11 different models implemented through the STOP program in Ontario between 2005 and 2016, via novel linkage of self-report with health administrative data. Treatment seeking smokers were similar across models on smoking characteristics. However, patients were not homogeneous on sociodemographic and health characteristics across models, highlighting the need to have a variety of treatment delivery options to adequately reach sub-populations who smoke in Ontario.

Though treatment protocols were not modified for each model in an effort to target different sub-populations, a number of factors may have contributed to the observed variation between STOP models. Enrollees in the Web, Telephone and Pharmacy models tended to be of generally higher socioeconomic status, lower comorbidity and younger age. Similarity between patients reached by these models may have been due to the use of similar mass media advertising strategies to promote awareness of the models; the use of different or additional advertising strategies to target specific subgroups (e.g., low-income smokers) may modify the demographic profile of patients seeking care via these models. These models also used similar web-based and telephone screening procedures. Internet and telephone access may have been limited in lower-income smokers [34–36], resulting in these models serving more socially advantaged smokers. Our findings are also consistent with prior evidence that individuals with higher levels of education are more likely to use telephone- and Internet-based smoking

**Table 3. Healthcare service utilization in the two years prior to enrollment, by STOP model.**

| Model | FHT | NPLC | CHC-B | AA | Tertiary | CHC-A | Workshop | PHU | Pharmacy | Web | Telephone |
|---|---|---|---|---|---|---|---|---|---|---|---|
| | N = 32,618 | N = 566 | N = 5,862 | N = 3,408 | N = 1,672 | N = 401 | N = 18,539 | N = 1,509 | N = 6,412 | N = 6,748 | N = 29,567 |
| Outpatient physician visits. No. (%), unless otherwise specified; No. visits reported as Median (IQR) | | | | | | | | | | | |
| Primary care | 31,077 (95.3) | 379 (67.0) | 3,645 (62.2) | 3,201 (93.9) | 1,580 (94.5) | 256 (63.8) | 16,684 (90.0) | 1,376 (91.2) | 5,835 (91.0) | 5,982 (88.6) | 27,055 (91.5) |
| No. visits | 6 (3–11) | 2 (0–7) | 2 (0–9) | 11 (5–22) | 11 (5–20) | 1 (0–6) | 7 (3–14) | 7 (3–15) | 7 (3–14) | 6 (2–12) | 8 (3–15) |
| Specialist | 22,482 (68.9) | 428 (75.6) | 4,286 (73.1) | 2,754 (80.8) | 1,312 (78.5) | 288 (71.8) | 12,487 (67.4) | 971 (64.3) | 3,851 (60.1) | 3,992 (59.2) | 17,920 (60.6) |
| No. visits | 2 (0–6) | 3 (1–9) | 3 (0–8) | 5 (1–13) | 4 (1–11) | 2 (0–7) | 2 (0–7) | 2 (0–5) | 1 (0–5) | 1 (0–5) | 1 (0–5) |
| MHA[a] | 12,920 (39.6) | 248 (43.8) | 2,331 (39.8) | 2,729 (80.1) | 1,000 (59.8) | 143 (35.7) | 8,329 (44.9) | 741 (49.1) | 2,696 (42.0) | 2,843 (42.1) | 13,086 (44.3) |
| No. visits | 0 (0–2) | 0 (0–5) | 0 (0–3) | 6 (1–19) | 1 (0–7) | 0 (0–1) | 0 (0–3) | 0 (0–3) | 0 (0–2) | 0 (0–2) | 0 (0–2) |
| Emergency department (ED) visits. No. (%), unless otherwise specified; No. visits reported as Median (IQR) | | | | | | | | | | | |
| Any | 20,568 (63.1) | 379 (67.0) | 3,803 (64.9) | 2,545 (74.7) | 1,028 (61.5) | 216 (53.9) | 11,134 (60.1) | 1,053 (69.8) | 3,542 (55.2) | 3,953 (58.6) | 15,433 (52.2) |
| No. visits | 1 (0–3) | 1 (0–3) | 1 (0–3) | 2 (0–5) | 1 (0–2) | 1 (0–2) | 1 (0–3) | 1 (0–3) | 1 (0–2) | 1 (0–2) | 1 (0–2) |
| MHA[b,c] | 2,193 (6.7) | 100 (17.7) | 767 (13.1) | 1,223 (35.9) | 179 (10.7) | 42 (10.5) | 1,570 (8.5) | 160 (10.6) | 425 (6.6) | 507 (7.5) | 1,966 (6.6) |
| Hospitalizations. No. (%), unless otherwise specified; No. visits reported as Median (IQR) | | | | | | | | | | | |
| Any[c] | 5,875 (18.0) | 107 (18.9) | 1,085 (18.5) | 755 (22.2) | 462 (27.6) | 58 (14.5) | 2,729 (14.7) | 278 (18.4) | 814 (12.7) | 936 (13.9) | 3,906 (13.2) |
| MHA[b,c] | 345 (1.1) | † | 105 (1.8) | 189 (5.5) | 50 (3.0) | 7 (1.7) | 232 (1.3) | 58 (3.8) | 88 (1.4) | 58 (0.9) | 534 (1.8) |
| Total healthcare cost (Canadian dollars) | | | | | | | | | | | |
| Median | 3,371 | 4,965 | 5,045 | 9,393 | 5,586 | 3,155 | 3,009 | 2,618 | 1,834 | 1,904 | 1,787 |
| (IQR) cost | (1,260–9,503) | (1,486–14,267) | (1,482–14,687) | (2,779–23,165) | (1,663–15,509) | (960–8,826) | (993–8,952) | (888–8,219) | (649–5,656) | (633–6,213) | (610–5,785) |

Percent (%) observed are within valid data (missing data ignored) unless otherwise indicated; AA = Addiction Agency, CHC = Community Health Centre,

FHT = Family Health Team, NPLC = Nurse Practitioner-Led Clinic, PHU = Public Health Unit, MHA = mental health and addictions;

† suppressed due to small cell counts (<6).

[a] MHA-related outpatient visits include primary care providers (family physicians and general practitioners) and psychiatrists (S3 Appendix).

[b] For definitions of MHA-related visits to emergency departments and hospitalizations, see S3 Appendix.

[c] Median (IQR) no. visits = 0 (0–0) for all models and therefore not shown.

cessation services [21]. In contrast, practitioner-mediated models in primary or tertiary care settings reached smokers who were more socially disadvantaged and/or had more complex healthcare needs and usage; these models, in which smoking cessation treatment is individualized to the needs of the patient, are well-suited to this more complex patient population [37, 38]. However, even within these settings, there was variation. The Addiction Agency, Community Health Centre-B and Nurse Practitioner-Led Clinic models reached patients with lower socioeconomic status and higher levels of comorbidity and healthcare utilization. Community Health Centres had the highest proportion of patients with low income, and highest levels of comorbidity, consistent with their mandate to serve patients facing barriers to accessing healthcare [39]. The Family Health Team model reached older patients with higher medical comorbidity, but not the most socially disadvantaged. These findings are in contrast with a previous comparison of patient demographics and case-mix of primary care models in Ontario [40], where it was reported that Family Health Teams have somewhat wealthier and healthier populations. Cessation patients in these models were relatively wealthy, but also experienced the poorer health status typical of smokers.

For most models the proportion of rural dwelling patients was close to or greater than the proportion of Ontarians residing in rural communities during the study timeframe

(14%-15%) [41, 42]. The Family Health Team model reached the largest number of rural residents, consistent with the overrepresentation of Family Health Teams in rural areas [39] and the large number of patients reached by this model. The earlier Community Health Centre-A and Public Health Unit models reached a higher proportion of rural patients, although this appeared to reflect the location of the small number of sites that partnered in these earlier models. Current models, operating through a larger number of Public Health Units (via the Workshop model) and Community Health Centre sites, have a lower proportion of rural residents. Although direct-to-smoker models were expected to have fewer barriers to access, particularly those related to geography, they reached a lower proportion of rural patients. The reasons are unclear but may be related to the presence of additional barriers in rural communities, such as lower levels of Internet access [35], income and education [43].

Our novel linkage study adds new information to what was previously known about Ontario smokers and their use of the healthcare system. Previous linkage studies have documented increased healthcare service utilization and costs among self-reported smokers in Ontario [40, 44], and demonstrated the effectiveness [45] and cost-effectiveness [46] of a hospital-based smoking cessation treatment program on subsequent healthcare service utilization and mortality. Our study has revealed variation in sociodemographic and health characteristics between smokers served by different treatment models, a comparison not previously reported. While these findings suggest a one-size-fits-all approach to treatment programs may not reflect patient heterogeneity, further work is needed to establish the optimal strategies for tailoring smoking cessation treatment programs for patient populations with different comorbid conditions [47].

Comparing the sociodemographic and health profile of smokers who sought treatment via each of the STOP models can help identify which models are able to reach more vulnerable groups. A disproportionate share of tobacco-related health burden is borne by already disadvantaged groups [48], such as those with lower socioeconomic status or comorbid mental illness, due to increased prevalence of tobacco use, lower cessation rates, barriers to accessing treatment and/or greater vulnerability to associated health risks [49]. Current findings indicate that not all models equally reached socially disadvantaged, and rural dwelling, individuals that traditionally experience greater barriers to accessing treatment. These findings have important implications for efforts to reduce and eliminate tobacco-related health disparities. Evidence from the UK's National Health Service smoking cessation services found that despite lower quit rates in disadvantaged areas, higher reach in these areas lead to a small net positive impact of the services on reducing health inequalities [50]. However, to further reduce inequity, additional research is needed to identify treatment strategies that are effective for individuals with various forms of disadvantage. We are developing and testing methods to address different co-morbidities during smoking cessation treatment in order to improve cessation outcomes (Clinicaltrials.gov registration numbers NCT03108144 and NCT03130998) [51, 52].

We note the importance of considering the context when interpreting findings from the current study, and before generalizing to other settings. Ontario has universal healthcare, and the STOP program, as part of that healthcare infrastructure, has been available to all patients regardless of personal financial means. In a setting where access to health care per se, and specific models of care, may vary based on financial means, comparisons between patient populations reached by different treatment models may reveal lesser or greater disparities or reflect contextual factors not measured in this study (e.g., ethnicity). Implementers in other settings are encouraged to consider and examine factors influencing disparity in their own local context.

## Limitations

A potential weakness of this study is selection bias owing to exclusion of patients who declined consent to link their data. Although only 15% of eligible enrollments declined consent for linkage overall, there was variation by model. Decline rate was 0% in the Web, Telephone, Pharmacy and Workshop models and minimal in the Tertiary Care, Public Health Unit and Community Health Centre-A models (<13%). However, the more recently implemented practitioner-mediated models had a higher decline rate (21–29% for the Community Health Centre-B, Nurse Practitioner-Led Clinic and Family Health Team models and 46% in the Addiction Agency model). Overall, records sent for linkage were older, more likely to be unemployed and of lower income, but did not differ on sex. Second, the results of these descriptive analyses do not imply concurrent choice by Ontarian smokers. The models have not been available all at once and patients served by earlier or later models might differ in ways related to time (e.g., changes in healthcare, Internet access, etc.). Further, in practitioner-mediated models, treatment is available only to patients served by individual practices that opted to implement the STOP program. Thus, patients did not have equal opportunity to access different models and comparisons presented are not intended to suggest that patient characteristics are associated with preference for one model over another. Moreover, findings may not generalize to an alternative combination of models.

## Conclusions

The current study found that persons seeking to quit smoking through a publicly funded smoking cessation treatment program in Ontario, though similar on smoking characteristics, varied on sociodemographic characteristics and health status according to the setting in which they were enrolled and received treatment. While practitioner-mediated models in specialized and primary care settings reach smokers with more complex healthcare needs that may require more medical supervision, alternative strategies and settings appear better suited to reach younger smokers before such comorbidities develop. These findings suggest that, in the interest of providing equitable access to smoking cessation treatment, the publicly funded healthcare system should ensure that some variation of these model types be available to all residents of Ontario who smoke.

## Supporting information

**S1 Appendix. Prevalent comorbidity algorithms and associated validation papers.**
(PDF)

**S2 Appendix. Population-based health administrative datasets used for cost ascertainment.**
(PDF)

**S3 Appendix. Mental health and addictions-related outcomes.**
(PDF)

## Acknowledgments

We thank the collaborating organizations, healthcare practitioners and Ontarians served by STOP. We thank Kinwah Fung for methodological support. Parts of this material are based on data and/or information compiled and provided by the Canadian Institute for Health Information (CIHI), the Immigration, Refugees and Citizenship Canada (IRCC), and Cancer Care Ontario (CCO). However, the analyses, conclusions, opinions and statements expressed herein

are those of the authors and not necessarily those of CIHI, IRCC, CCO or the Ontario Ministry of Health and Long-Term Care.

## Author Contributions

**Conceptualization:** Dolly Baliunas, Laurie Zawertailo, Susan J. Bondy, Peter L. Selby.

**Data curation:** Dolly Baliunas.

**Formal analysis:** Longdi Fu.

**Funding acquisition:** Dolly Baliunas, Laurie Zawertailo, Peter L. Selby.

**Investigation:** Laurie Zawertailo, Peter L. Selby.

**Methodology:** Dolly Baliunas, Laurie Zawertailo, Peter L. Selby.

**Project administration:** Sabrina Voci, Evgenia Gatov.

**Resources:** Laurie Zawertailo.

**Supervision:** Dolly Baliunas, Peter L. Selby.

**Visualization:** Dolly Baliunas, Sabrina Voci, Evgenia Gatov.

**Writing – original draft:** Dolly Baliunas, Sabrina Voci, Evgenia Gatov.

**Writing – review & editing:** Dolly Baliunas, Laurie Zawertailo, Sabrina Voci, Evgenia Gatov, Susan J. Bondy, Longdi Fu, Peter L. Selby.

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
