## [Decision Letter · Decision Letter 0]

24 Apr 2020

PONE-D-20-07364

Variability in patient sociodemographics, clinical characteristics, and healthcare service utilization among 107,302 treatment seeking smokers in Ontario: A cross-sectional comparison

PLOS ONE

Dear Dr. Baliunas,

Thank you for submitting your manuscript to PLOS ONE. After careful consideration, we feel that it has merit but does not fully meet PLOS ONE’s publication criteria as it currently stands. Therefore, we invite you to submit a revised version of the manuscript that addresses the points raised during the review process.

We would appreciate receiving your revised manuscript by Jun 08 2020 11:59PM. To enhance the reproducibility of your results, we recommend that if applicable you deposit your laboratory protocols in protocols.io, where a protocol can be assigned its own identifier (DOI) such that it can be cited independently in the future. For instructions see: http://journals.plos.org/plosone/s/submission-guidelines#loc-laboratory-protocols

We look forward to receiving your revised manuscript.

Kind regards,

Stanton A. Glantz

Academic Editor

PLOS ONE

2. Please ensure you have included the registration number for the clinical trial referenced in the manuscript.

"I have read the journal's policy and the authors of this manuscript have the following competing interests: DB reports grants from the Canadian Cancer Society Research Institute during the conduct of the study; she also reports grants from a Pfizer GRAND award, outside the submitted work. LZ reports grants from the Canadian Cancer Society Research Institute during the conduct of the study; she also reports grants from Pfizer GRAND Awards, outside the submitted work. PLS reports receiving grants from the Centre for Addiction and Mental Health, Ontario Ministry of Health and Long-Term Care, Health Canada, Canadian Institutes of Health Research, Canadian Cancer Society Research Institute, Canadian Partnership Against Cancer, Medical Psychiatry Alliance, Ontario Neurotrauma Foundation, the Patient-Centered Outcomes Research Institute, Bhasin Consulting Fund Inc and Pfizer Inc./Canada. PLS also reports serving on advisory boards and receiving consulting fees and honoraria from Pfizer Canada Inc., Johnson & Johnson Group of Companies, Evidera Inc., Miller Medical Communications, NVision Insight Group, Myelin & Associates, University of Ottawa Heart Institute, Royal College of Physicians and Surgeons of Canada, Royal Victoria Regional Health Centre, the University of Toronto Department of Family and Community Medicine, Northern Ontario School of Medicine, Canadian Partnership Against Cancer, Battle River Treaty 6 Healthcare, Lung Association of Nova Scotia, Exchange Summit, Toronto Public Health, Ontario Association of Public Health Dentistry and ECHO. PLS also reports that MedPlan Communications assisted in organizing Pfizer Canada Inc. Advisory Board events for which he was a consultant and that Pfizer Inc., Novartis, and Johnson & Johnson are vendors of record for providing free/discounted smoking cessation pharmacotherapy for research studies on which he is principal or co-investigator. No other author has competing interests to declare."

Reviewers' comments:

Reviewer's Responses to Questions

**Comments to the Author**

1. Is the manuscript technically sound, and do the data support the conclusions?

Reviewer #1: Yes

Reviewer #2: Yes

2. Has the statistical analysis been performed appropriately and rigorously? 

Reviewer #1: Yes

Reviewer #2: Yes

3. Have the authors made all data underlying the findings in their manuscript fully available?

Reviewer #1: No

Reviewer #2: Yes

4. Is the manuscript presented in an intelligible fashion and written in standard English?

Reviewer #1: Yes

Reviewer #2: Yes

5. Review Comments to the Author

Reviewer #1: This is a well-written paper about the STOP smoking cessation program in Ontario, Canada and its 11 specific sub-programs. The manuscript describes the variability in patient sociodemographic and clinical characteristics, and in healthcare service utilization.

Comments are in order of appearance.

Around line 15 in the introduction, please consider adding that it has been shown that NRT and counselling offered together are more effective than NRT alone.

When mentioning all the different data sources, starting in line 67, please include references to all of them, not just the ones you list in Appendix 2. If you do not want to include the information in the references, then just add them to Appendix 2. Some readers might want to follow up.

Footnotes to Table 2: “** suppressed due to small cell count (<6)” should not be a category between * and *** which are about the proportions of missing values. How about giving the small cell count an entirely different symbol?

Please spell out MHA the first time it appears in the text, it is only spelled out in one of the table legends.

The manuscript has a lot of detail about the different Ontario programs and about the characteristics of those enrolled in them. This information is valuable for future smoking cessation programs trying to reach different populations of smokers in this very specific region. What would add to the value of the paper would be somewhat more of a discussion of how this might (or might not) be generalizable outside of Ontario. Can you make any recommendations for readers who are not in Ontario or not even in Canada?

Figures: Figures should be more self-explanatory. Please consider the full names of the 11 sub-programs in the legend, including the years in which they were active.

Reviewer #2: This is an interesting and well-written manuscript describing sociodemographic, clinical and healthcare services utilization among smokers seeking smoking cessation services in Ontario Canada. The authors found that there were differences in the case-mix of populations that were reached by different cessation models, but less clear were the implication of these findings and how they could promote equity in access to cessation programs. I highlight some opportunities for clarifications below.

Abstract:

- The abstract would be clearer if the authors described in the background the different models of cessation care that are available to patients through the STOP program. While a long description is not necessary, a short description describing general approaches such as in-person, telephone etc. would be helpful.

- It is also not clear how the authors came up with the 107,302 case mix. For e.g., is this from a primary care populations and what clinics/hospitals were represented in this population. A brief description of the fact that the 107,302 people represent people who engaged in the STOP program would be helpful.

- The implications could be made clearer in terms of models that may be most effective for subpopulations with specific comorbidities, versus a more general population-based strategy to address tobacco use

Introduction

- The introduction is really clear in terms of its objective.

- My only suggestion is to provide a brief description of the 11 models or at least refer the reader to the Table and say that some were initiated by a referral (??) and some were self-referred. And who they were intended to target at the time of inception. i.e., were some models specifically more intensive because they were seeing a higher risk population (e.g., people in addition treatment). While some of this is described in the methods, I might bring some of this info up into the intro or at least refer the reader to the Table 1. I think some clarity on these models in the text would really help because when you get the results, it gets very confusing to keep all the models straight.

Methods

- Why was a long-term costs approach not considered in this study, i.e., costs past the index visit?

- Do the different STOP models for cessation, self-select for specific populations. For e.g., the web-based model may be best directed to a group that may not be very sick, whereas the addiction treatment model may target a higher needs group. If so, then costs would be vastly different based on the population seeking cessation treatment from a specific model. What information is cost adding in this analysis?

- How was missing data treated?

Results

- The most difficult part of the results is keeping the models straight and simplifying the results in the text so that it’s clear what the primary characteristics are for each of the models. One suggestion is to not present as many characteristics in the text but only emphasize the key ones so that the reader comes out with some key findings.

- Similarly Table 2 is really long and has a lot of information. Could you divide this into two tables?

- Also the sample sizes for some of the models are quite small, example CHC or nurse practitioner led clinics – is there utility in combining some of these smaller models? Or is the reasoning to keep them separate because they were intended to target specific populations.

Discussion

- The discussion is well-written. The main questions that I have that can be addressed in the discussion:

o whether these models catered to specific sub-populations at the outset and if that has changed over time.

o Are there cessation outcomes data available for patients enrolled in each of the STOP models. If there are, do the cessation rates reflect characteristics of the population seeking specific models?

o How much is reach defined by what is currently available and not what can be made available? i.e., if there were more access to web-based/telephone models in rural setting would there be equal reach compared to the NP based models. What can this study inform us about models that might be most impactful for low-income populations in Ontario.

6. PLOS authors have the option to publish the peer review history of their article (what does this mean?). If published, this will include your full peer review and any attached files.

Reviewer #1: No

Reviewer #2: No

---

## [Author Response · Author response to Decision Letter 0]

3 Jun 2020

Please see the attached Response to Reviewers document.

---

## [Editor Report · Decision Letter 1]

16 Jun 2020

PONE-D-20-07364R1

Variability in patient sociodemographics, clinical characteristics, and healthcare service utilization among 107,302 treatment seeking smokers in Ontario: A cross-sectional comparison

PLOS ONE

Dear Dr. Baliunas,

Thank you for submitting your manuscript to PLOS ONE. After careful consideration, we feel that it has merit but does not fully meet PLOS ONE’s publication criteria as it currently stands. Therefore, we invite you to submit a revised version of the manuscript that addresses the points raised during the review process.

You have done a nice job of responding to the reviewers' comments.

I only have one small request.  In the conclusion of the abstract you have vague statements that the healthcare system should ensure "some variation" in the cessation models used depending on patient and context.  The paper would be a lot better if you can make these statements more specific and concrete.  You have the relevant information in the Discussion.

We look forward to receiving your revised manuscript.

Kind regards,

Stanton A. Glantz

Academic Editor

PLOS ONE

---

## [Author Response · Author response to Decision Letter 1]

18 Jun 2020

We have revised the Discussion section of the Abstract to address the requested change, and made some further revision to the remainder of the Abstract in order to meet the word count limit.

---

## [Editor Report · Decision Letter 2]

22 Jun 2020

Variability in patient sociodemographics, clinical characteristics, and healthcare service utilization among 107,302 treatment seeking smokers in Ontario: A cross-sectional comparison

PONE-D-20-07364R2

Dear Dr. Baliunas,

We’re pleased to inform you that your manuscript has been judged scientifically suitable for publication and will be formally accepted for publication once it meets all outstanding technical requirements.

I have passed on your request to list Dr. Selby's email in the paper.  If you don't get an answer at this point, I suggest that you raise the question with the production staff as the manuscript moves forward into print.

Kind regards,

Stanton A. Glantz

Academic Editor

PLOS ONE
---

## [Editor Report · Acceptance letter]

29 Jun 2020

PONE-D-20-07364R2 

Variability in patient sociodemographics, clinical characteristics, and healthcare service utilization among 107,302 treatment seeking smokers in Ontario: A cross-sectional comparison 

Dear Dr. Baliunas:

I'm pleased to inform you that your manuscript has been deemed suitable for publication in PLOS ONE. Congratulations! Your manuscript is now with our production department. 

Kind regards, 

on behalf of

Professor Stanton A. Glantz 

Academic Editor

PLOS ONE